# SufiSent - Universal Sentence Representations Using Suffix Encodings

**Siddhartha Brahma**
IBM Research, Almaden, USA
`brahma@us.ibm.com`

## Abstract

Computing universal distributed representations of sentences is a fundamental task in natural language processing. We propose a method to learn such representations by encoding the suffixes of word sequences in a sentence and training on the Stanford Natural Language Inference (SNLI) dataset. We demonstrate the effectiveness of our approach by evaluating it on the SentEval benchmark, improving on existing approaches on several transfer tasks.

## 1 Introduction

In natural language processing, the use of distributed representations has become standard through the effective use of word embeddings. In a wide range of NLP tasks, it is beneficial to initialize the word embeddings with ones learnt from large text corpora like word2vec Mikolov et al. (2013) or GLoVe Pennington et al. (2014) and tune them as a part of a target task e.g. text classification. It is therefore a natural question to ask whether such standardized representations of whole sentences that can be widely used in downstream tasks, is possible.

There are two classes of approaches to this problem. Taking cue from word2vec, an unsupervised learning approach is taken by SkipThought Kiros et al. (2015) and FastSent Hill et al. (2016). More recently, the work of Conneau et al. (2017) takes a supervised learning approach. They train a sentence encoding model on the Stanford Natural Language Inference (SNLI) dataset Bowman et al. (2015) and show that the learnt encoding transfers well to to a set of transfer tasks encapsulated in the SentEval benchmark. This is reminiscent of the approach taken by ImageNet Deng et al. (2009) in the computer vision community.

One of the most effective ways of encoding a sentence $\mathbf{s}$ is to pass it through a recurrent neural network like a LSTM Hochreiter & Schmidhuber (1997) and use the last hidden state or a combination of the intermediate hidden states. Each intermediate state of a LSTM represents an encoding of a prefix of $\mathbf{s}$. In a bidirectional LSTM, an additional network is used to encode the prefixes of the reversed sequence of words. Although this is equivalent to encoding the suffixes of $\mathbf{s}$, the suffixes are encoded in a direction reverse of the prefixes.

In this paper, we argue that encoding the suffixes of $\mathbf{s}$ in the forward direction can lead to better universal sentence representations. By Max-pooling the encodings of the prefixes and suffixes, we define a new sentence encoding that is trained on the SNLI dataset. We show through numerical experiments that the learned encodings improve upon existing supervised approaches on the SentEval benchmark. We call our suffix based sentence encoding model SufiSent.

## 2 Suffix Based Models

Let $\mathbf{s}$ be a sentence with $n$ words. We will use $\mathbf{s}[i\!:\!j]$ to denote the sequence of words from $\mathbf{s}[i]$ to $\mathbf{s}[j]$, where $i$ maybe less than $j$. Let $\vec{L}_p$ represent a LSTM (or any other RNN) that encodes *prefixes* of $\mathbf{s}$ in the forward direction. For the $i$-th word, we have

$$\vec{h}_{p,i} = \vec{L}_p(\mathbf{s}[1\!:\!i]) \tag{1}$$

Let $\vec{L}_s$ represent a LSTM that encodes *suffixes* of $\mathbf{s}$ in the forward direction.

$$\vec{h}_{s,i} = \vec{L}_s(\mathbf{s}[i\!:\!n]) \tag{2}$$

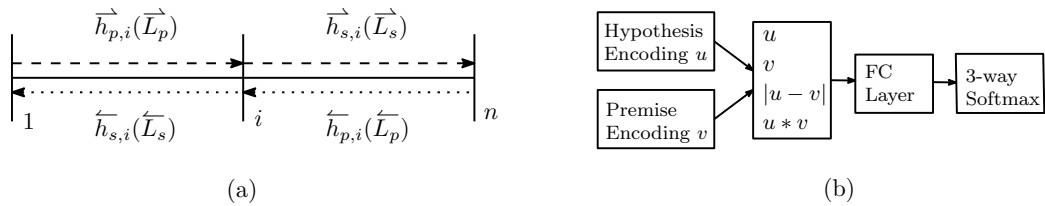

Figure 1: (a) Schematics of SUFISENT terms (b) Training architecture on SNLI dataset

Note that the $\vec{h}_{p,i}$ can be computed in a single pass over **s**, while computing $\vec{h}_{s,i}$ needs a total of $n$ passes over progressively smaller suffixes of **s**. As in bidirectional LSTMs, we also consider $\overleftarrow{L}_p$ and $\overleftarrow{L}_p$ that encodes the prefixes and suffixes of **s** in the backward direction.

$$\overleftarrow{h}_{s,i} = \overleftarrow{L}_s(\mathbf{s}[i\!:\!1]), \quad \overleftarrow{h}_{p,i} = \overleftarrow{L}_p(\mathbf{s}[n\!:\!i]), \tag{3}$$

Note that $\vec{h}_{p,i}$ encodes the same subsequence as $\vec{h}_{s,i}$, but in different directions. See Fig.1(a) for a schematic illustration. Let $d$ be the dimension of the hidden state of each of $\vec{L}_p, \vec{L}_s, \overleftarrow{L}_p, \overleftarrow{L}_s$. We consider the following sentence encodings.

- **SUFISENT** - We max-pool the $\vec{h}_{p,i}$ over all $i \in [1:n]$ to obtain $\vec{h}_p$ and max-pool the $\vec{h}_{s,i}$ to obtain $\vec{h}_s$. Similarly, we obtain $\overleftarrow{h}_p$ and $\overleftarrow{h}_s$ by max-pooling over $\overleftarrow{h}_{p,i}$ and $\overleftarrow{h}_{s,i}$ respectively. The final encoding is a concatenation of $\max(\vec{h}_p, \overleftarrow{h}_s)$ and $\max(\overleftarrow{h}_p, \vec{h}_s)$. The sentence encoding is of size $2d$. In contrast, a BiLSTM-Max model is a concatenation of $\vec{h}_p$ and $\overleftarrow{h}_p$.

- **SUFISENT-TIED** - This is same as above, but the weights of $\vec{L}_p$ and $\vec{L}_s$ are shared or tied. Similarly, the weights of $\overleftarrow{L}_p$ and $\overleftarrow{L}_s$ are tied. The sentence encoding is of size $2d$.

- **SUFISENT-CAT** - Similar to SUFISENT, we compute $\vec{h}_p, \vec{h}_s, \overleftarrow{h}_p, \overleftarrow{h}_s$. The sentence encoding is the concatenation of these four vectors and is of size $4d$.

- **SUFISENT-CAT-TIED** - This is same as SUFISENT-CAT, except the weights of $(\vec{L}_p, \vec{L}_s)$ and $(\overleftarrow{L}_p, \overleftarrow{L}_s)$ are tied. The size of the encoding is $4d$.

We use the SNLI dataset as the supervised dataset to train the encodings. SNLI is a large scale labelled dataset consisting of pairs of sentences (premise and hypothesis) and each pair is labeled by one of three labels - entailment, contradiction and neutral. As shown in Fig. 1(b), for each of the SUFISENT models, the encodings of the premise and hypothesis sentences are computed as $u$ and $v$. Following Mou et al. (2016), a feature vector consisting of $u$, $v$, $|u - v|$ and $u * v$ is fed into a fully connected layer(s), before computing the 3-way softmax in the classification layer.

## 3 TRAINING AND RESULTS

The encodings defined by SUFISENT and SUFISENT-TIED are trained on the SNLI dataset for the LSTM hidden dimensions $d \in \{256, 512, 1024, 2048\}$. The SUFISENT-CAT and SUFISENT-CAT-TIED encodings are trained for $d \in \{128, 256, 512, 1024\}$. This corresponds to sentence encoding dimensions of $512, 1024, 2048, 4096$ respectively. The FC layer has two layers of $512$ dimensions each. For optimization, we use SGD with an initial learning rate of 0.1 which is decayed by 0.99 after every epoch or by 0.2 if there is a drop in the validation accuracy. Gradients are clipped to a maximum norm of 5.0.

We evaluate the sentence encodings using the SentEval benchmark Conneau et al. (2017). This benchmark consists of 6 text classification tasks (MR, CR, SUBJ, MPQA, SST, TREC), one task on paraphrase detection (MRPC) and one on entailment classification (SICK-E). All these 8 tasks have accuracy as their performance measure. There are two tasks (SICK-R and STS14) for which the performance measure is Pearson and Pearson/Spearman correlation respectively. The trained encoding models are used to generate initial representations for the sentences in the transfer tasks, which are then tuned further. For more details, please refer to the above paper.

| Model | dim | SNLI dev | SNLI test | Transfer micro | Transfer macro |
|---|---|---|---|---|---|
| BiLSTM-Mean | 4096 | 79.0 | 78.2 | 83.1 | 81.7 |
| Inner-attention | 4096 | 82.3 | 82.5 | 82.1 | 81.0 |
| HConvNet | 4096 | 83.7 | 83.4 | 82.0 | 80.9 |
| BiLSTM-Max. | 4096 | **85.0** | 84.5 | 85.2 | 83.7 |
| SufiSent-Tied | 4096 | 84.7 | **84.6** | **86.6** | **85.1** |
| SufiSent | 4096 | 84.9 | 84.3 | 86.5 | 85.0 |
| SufiSent-Cat-Tied | 4096 | 84.8 | 84.4 | 86.2 | 84.5 |
| SufiSent-Cat | 4096 | 84.6 | 84.2 | 86.3 | 84.3 |

Table 1. Performance of SUFISENT* models on the SNLI dataset and on the validation sets of transfer tasks with accuracy as performance. Numbers in first four rows are taken from Conneau et al. (2017).

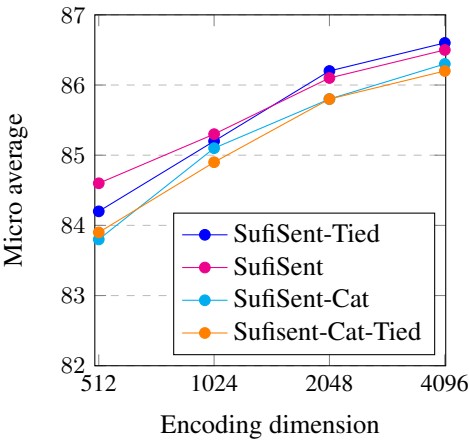

Figure 2: Scaling of micro average of accuracies on 8 tasks with encoding dimension.

| Model | MR | CR | SUBJ | MPQA | SST | TREC | MRPC | SICK-R | SICK-E | STS14 |
|---|---|---|---|---|---|---|---|---|---|---|
| *Unsupervised Training - Ordered Sentences* | | | | | | | | | | |
| FastSent | 70.8 | 78.4 | 88.7 | 80.6 | - | 76.8 | 72.2/80.3 | - | - | .63/.64 |
| SkipThought | 76.5 | 80.1 | 93.6 | 87.1 | 82.0 | 92.2 | 73.0/82.0 | 0.858 | 82.3 | .29/.35 |
| SkipThought-LN | 79.4 | 83.1 | 93.7 | 89.3 | 82.9 | 88.4 | - | 0.858 | 79.5 | .44/.45 |
| *Supervised Training on SNLI* | | | | | | | | | | |
| InferSent | 79.9 | 84.6 | 92.1 | 89.8 | 83.3 | **88.7** | 75.1/82.3 | 0.885 | **86.3** | .68/.65 |
| SufiSent | 80.3 | 84.7 | **92.8** | 90.1 | **83.4** | 88.0 | **75.4/82.9** | 0.886 | 85.7 | **.69/.66** |
| SufiSent-Tied | **80.6** | **85.4** | 92.2 | **90.3** | 83.1 | 88.4 | 74.3/82.3 | **0.887** | **86.3** | .68/.66 |
| SufiSent-Cat | 80.3 | 84.4 | 92.2 | 90.2 | 81.4 | 85.2 | 74.6/82.5 | 0.883 | 86.0 | .66/.63 |
| SufiSent-Cat-Tied | 79.8 | 84.8 | 92.3 | 90.2 | 82.3 | 86.6 | 74.5/82.5 | 0.880 | 85.8 | .64/.61 |
| *Supervised Training on AllNLI* | | | | | | | | | | |
| InferSent | 81.1 | 86.3 | 92.4 | 90.2 | 84.6 | 88.2 | 76.2/83.1 | 0.884 | 86.3 | .70/.67 |

Table 2. Test set performance over the transfer tasks in SentEval. For MRPC, we report accuracy and F1 score. The dimension for the SUFISENT* and Infersent models is 4096. All numbers except for our models are taken from Hill et al. (2016) and Conneau et al. (2017).

As can be seen from Table 1 and Fig. 2, among the models proposed in this paper, the SUFISENT-TIED model with dimension 4096 has the best test accuracy on the SNLI dataset and also the best macro and micro average of the validation set accuracies in the 8 transfer tasks identified above. It also performs significantly better than the BiLSTM-Max (InferSent) of Conneau et al. (2017), which only uses the max of the prefix encodings in both directions. The performance steadily improves with increasing encoding dimension, as shown in Fig. 2. The test set performance of SUFISENT-TIED on SNLI improves on InferSent too. SUFISENT and SUFISENT-TIED are close, with the latter edging forward in higher dimensions.

Table 2 compares the test set performance of the SUFISENT models with InferSent on each of the transfer tasks for the encoding dimension of 4096. For the same training set (SNLI), both SUFISENT-TIED and SUFISENT improves or matches InferSent on 7 of the 10 tasks. The improvement is particularly significant for MR, CR, SUBJ and MPQA. The performance of models trained on unlabeled data and the InferSent model on the larger AllNLI dataset is also shown for comparison.

To conclude, we propose SUFISENT - a new universal sentence encoding that is computed by max-pooling over the encodings of the suffixes and prefixes of sentences, in both the forward and backward directions. Preliminary results obtained by training on the SNLI dataset shows promise, improving over existing approaches on many transfer tasks in the SentEval benchmark. In future work, we plan to train SUFISENT on the larger AllNLI dataset, explore its use in other NLP tasks as a basic representation primitive and address computational efficiency issues.

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
