# OpenReview forum: "SufiSent - Universal Sentence Representations Using Suffix Encodings"
_ICLR.cc/2018/Workshop — Accept_

### Official Review · AnonReviewer3 · 2018-03-09
**Interesting incremental work**

**Rating:** 6
**Confidence:** 4

**Review:**

The paper proposes a sentence representation method by encoding the suffixes of word sequence in a sentence with BiLSTMs. In the forward direction, the proposed model encodes both prefixes and suffixes. To show the effectiveness, the SufiSent models were trained and tested on SNLI and a transfer SentEval benchmark and achieved better performance than the baseline, BiLSTM-Max (Conneau et al. 2017). The paper compares several variants of SufiSent such as tied parameters and concatenation of vectors.

The paper proposes an interesting extension (though novelty is rather limited) to the original BiLSTM encoders. Content of the paper is well structured and easy to follow.

Pros:
1. This is an interesting extension to the original BiLSTM encoders.
2. Experiments show some improvement on the SNLI and the transfer tasks.

Cons:
1. Some more analysis and discussion on the benefit of encoding suffixes would help readers understand the emperical improvement.
2. The suffix encoding needs a total of N passes over progressively smaller suffixes of sentence. It may have a higher computation cost than the baseline. It would be interesting to provide some comparison on speed between baselines and the proposed models.

---

### Official Review · AnonReviewer2 · 2018-03-10
**Minor contribution with mixed results**

**Rating:** 6
**Confidence:** 3

**Review:**

This paper proposes to extend the bi-directional LSTM (BiLSTM) for suffixes encoding. In sum, this paper is quite clear, although I would like to see more motivation for the proposed approach. In my opinion, the only difference of this model compared to the BiLSTM is that the author presents information encoded in each point of a sentence twice (left-to-right and right-to-left context). It means also that the number of parameters is doubled compared to a traditional BiLSTM. Therefore, in order to have a fair comparison, the author should double the number of parameters of the BiLSTM too (Table 1).

In Table 2, the results are mixed. The author should provide more intuition and insights to readers to have a better understanding why. Furthermore, I suggest the author to add a BiLSTM with self-attention as another baseline system.

---

### Official Review · AnonReviewer1 · 2018-03-13
**Solid focused contribution**

**Rating:** 7
**Confidence:** 4

**Review:**

This paper presents a simple modification to the standard BiLSTM architecture for sentence classification that results in increased performance over a number of tasks. The basic idea is to encode sentence suffixes using multiple passes through an LSTM (rather than just a single pass as in the backwards side of a BiLSTM). While the gains are quite small over the InferSent baseline on most tasks, overall I think the results are good enough to make NLP researchers think about trying it out to squeeze out some extra accuracy (although the computational inefficiency might make them think twice! it would be good to mention the training time in the paper). The paper does not attempt to explain *why* these suffix representations are better than standard BiLSTM ones, although three pages is not much room; nevertheless, I wonder if looking at examples for which your model is correct but InferSent is wrong would be informative.

Additionally, it would be nice to see mean/variance across multiple runs reported in the results tables, as there is really no reason to only run one experiment on these small datasets, and the reported accuracy differences are often miniscule.

Overall, I would lean towards accepting this paper, as I think its empirical contribution is appropriate for a workshop submission despite an analysis of why the method works.

---

### Public Comment · ~Samuel_R._Bowman1 · 2018-03-06
**Interesting result**

Two questions:
– Have you tried training plain InferSent using your same training setup and the same number of parameters? There's always the risk with experiments like these that your results will have more to do with subtle training decisions like early stopping criterion or initialization than with the target change.
– What do you think suffix encoding gets you? Is there some useful strategy for language understanding that's easier to learn with a suffix encoder than with a plain BiLSTM?

---

> ### Public Comment · ~Siddhartha_Brahma1 · 2018-03-07
> **Regarding the experiments and language understanding from SufiSent**
>
>
> Thank you for reading the paper. Both the questions are very pertinent.
>
> 1. We use exactly the same training setup as used by the InferSent paper - namely SGD with initial learning rate of 0.1, which is decayed by 0.99 after every epoch if the validation accuracy improves over the current best or by 0.2 if the validation accuracy does not improve. The training is continued until the learning rate drops below 1e-5. The word embeddings are initialized using GloVe vectors and are not updated during training. We also use the same classification layer to make the results comparable (one fully connected layer of dimension 512; there is a typo in our paper submission which incorrectly mentions this as two). We also retrained the InferSent model in our setup, obtaining results comparable to the ones published in the paper, hence we decided to stick with the numbers from the paper.
>
> As for the number of parameters, a SufiSent-Tied model (which uses the same LSTM for the prefixes and suffixes in one direction) has exactly the same number of parameters as a BiLSTM model with the same LSTM encoding dimension. In fact, the SufiSent-Tied is also our best overall model (see Table 1). A SufiSent model, which uses two different LSTMs for the prefixes and suffixes, has double the number of parameters. As shown in Figure 2, SufiSent has an upper hand over SufiSent-Tied at lower dimensions, but the latter becomes better with the increased capacity of the LSTMs at higher dimensions.
>
> In summary, we have taken care to make our results as comparable as possible to the InferSent results and the SufiSent models show improvement over InferSent under the same training conditions.
>
> 2. This is a very good question, and we will try to give our best understanding of why encoding suffixes is beneficial.
> One way to think about the intermediate states in a BiLSTM encoder is to see each state as a distinct “encoded view” of the sentence. However, the fact that the suffixes are encoded in a reverse order, makes them not directly compatible with the prefix encodings. By encoding the suffixes, we can directly combine the prefix and suffix encodings (using a max in SufiSent), thereby giving richer encoded views.
>
> Perhaps a more convincing reason is the following. The $k$-th intermediate state in a LSTM encoding is constrained to be a function of the $k-1$-th state and the $k$-th word. What SufiSent is doing is resetting the state at each word and forcing the LSTM to “learn from scratch”. Another way to put it would be to think of SufiSent as a single LSTM being repeatedly fed a sentence with larger and larger prefixes dropped out, and then their final encodings being combined by max-pooling. We believe this gives it a better generalization ability as evidenced by the better results compared to a BiLSTM-Max/InferSent encoder on transfer tasks in SentEval.
>
> In terms of language understanding, one important by-product of computing encodings for each suffix is that we have encodings for each subsequence of a sentence. If understanding a sentence implies identifying key parts and discarding others, these subsequence encodings could be combined in interesting ways to come up with an encoding of a sentence that is able to retain specific parts of the sentence, in a task specific manner. On the other hand, external knowledge like a parse tree could be used to combine the subsequence encodings, thereby producing a model with a strong inductive bias. We are actively investigating all these aspects of SufiSent.

---

### Decision · Program_Chairs · 2018-03-20
**ICLR 2018 Workshop Acceptance Decision**

**Decision:**

Accept

**Comment:**

Congratulations, your paper was accepted to the ICLR workshop.